# Pressure Reduction Enhancing the Production of 5-Hydroxymethylfurfural from Glucose in Aqueous Phase Catalysis System

**DOI:** 10.3390/polym13132096

**Published:** 2021-06-25

**Authors:** Ke Ke, Hairui Ji, Xiaoning Shen, Fangong Kong, Bo Li

**Affiliations:** 1State Key Laboratory of Pulp and Paper Engineering, South China University of Technology, 381 Wushan Road, Guangzhou 510640, China; 201820126057@mail.scut.edu.cn (K.K.); mssxn@mail.scut.edu.cn (X.S.); 2School of Light Industry Science and Engineering, Qilu University of Technology (Shandong Academy of Sciences), Jinan 300175, China; jihairui@yeah.net; 3State Key Laboratory of Biobased Material and Green Papermaking, Qilu University of Technology (Shandong Academy of Sciences), Jinan 300175, China; kfgwsj1566@163.com

**Keywords:** glucose, 5-hydroxylmethyfurfural, aqueous phase, heterogeneous catalysts, pressure reduction

## Abstract

5-hydroxymethylfurfural (HMF) obtained from biomass is an important platform chemical for the next generation of plastics and biofuel production. Although industrialized, the high yield of HMF in aqueous systems was rarely achieved. The main problem is that HMF tends to form byproducts when co-adsorbed with water at acid sites. In this study, the pressure was reduced to improve the maximum yield of HMF from 9.3 to 35.2% (at 190 °C in 60 min) in a glucose aqueous solution. The mechanism here involved water boiling as caused by pressure reduction, which in turn promoted the desorption of HMF from the solid catalyst, thereby inhibiting the side reaction of HMF. Furthermore, the solid catalysts could be reused three times without a significant loss of their catalytic activity. Overall, this work provides an effective strategy to improve the yield of HMF in water over heterogeneous catalysts in practice.

## 1. Introduction

Biomass is the most abundant renewable resource in the world, with the advantages of low pollution, wide distribution, and renewability [1]. The development and utilization of biomass is an effective way to alleviate global energy problems [2,3,4,5,6]; in addition to this aspect, some polymers can be prepared from biomass-based platform compounds. The interest in the preparation of polymeric materials based on biomass resources is growing. Undoubtedly, polyethylene terephthalate (PET) is the one of the largest volume industrial polymers. However, PET is not easy to degrade, which in turn causes environmental pollution and can damage human health. Recently, researchers designed a biodegradable plastic (PEF) synthesized by replacing terephthalic acid with furan-2,5-dicarboxylic acid (FDCA) for replacing PET [7,8,9,10]. FDCA, the oxidation product of HMF, is the key point to produce PEF [11,12,13,14,15]. Therefore, it is very necessary to explore HMF in order to produce FDCA for furthering the polymeric industry.

There are many raw materials for the production of HMF such as glucose, fructose, sucrose, and cellulose. Literature has indicated that it is easiest to produce HMF for fructose [16,17], but glucose should be the most suitable raw material to prepare HMF, as it is a cheaper and more abundant material. The glucose–HMF conversion path involved isomerizing glucose to fructose over a Lewis acid, followed by dehydrating fructose to form HMF with Brønsted acid [18,19,20]. It has been reported in literature that homogeneous catalysts could effectively catalyze glucose to obtain HMF [21,22,23,24,25,26]. However, the disadvantages of homogeneous catalysts, including corrosion, environmental pollution, and difficulty of recycling, limit their application and spur the development of heterogeneous catalysts [27,28,29,30]. Moreover, solvents such as organic solvents, biphasic systems, and ionic liquid are the focus in the HMF production [31,32,33,34,35], but they have many problems. DMSO, a typical organic solvent, is hard to separate from HMF because of it having a high boiling point and it being expensive and environmentally unfriendly. Biphasic systems cannot avoid the use of toxic solvents, and the viscosity of the bulk of ionic liquids is too high to mass transfer in reactions. Water has the properties of a low boiling point, low viscosity, and high environmental protection. The use of heterogeneous catalysts to catalyze biomass to produce HMF in water meets the requirements of industrial production such as environmental protection and the easy separation of products.

Zirconia (ZrO_2_) was reported to possess acid properties, which makes it a good choice as a catalyst and catalyst support [36]. For the first time, ZrO_2_ was used for catalyzing glucose to produce HMF in water, with the yield being 10.0% [37]. Chareonlimkun et al. [38] revealed that the mechanism of using sulfated zirconia (SO_4_–ZrO_2_) can obtain a higher HMF yield than ZrO_2_. The reason is that ZrO_2_ acts as Lewis acid sites to promote the isomerization of glucose to fructose [39], and Brønsted acid sites introduced by acidification further promote the dehydration of fructose to HMF [40]. It was indicated that the yield of HMF was less than 30.0% from glucose in water. The main reason is that HMF is an unstable intermediate that is prone to side reactions [41,42,43]. Side reactions mainly include two routes: (a) the rehydration reaction between HMF and water is to produce organic acids such as levulinic acid and formic acid, and (b) the condensation reaction between HMF and sugars or other products is to form soluble polymer or insoluble humins. Hence, suppressing the occurrence of side reactions is an important method for increasing the yield of HMF.

There are very few reports about inhibiting the unwanted reactions of HMF in water. Recently, a method to increase the yield of HMF by adding a low-boiling solvent to the water–ionic liquid system was discovered [34]. The research considered how the bubbles produced by boiling promoted the rate of interphase mass-transfer of HMF in reactions, which increased the extraction ratio of HMF in ionic liquid and avoided the side reactions of HMF.

Based on the inspiration of this theory, this work proposes a method for pressure reduction to promote the production of HMF from glucose with a heterogeneous catalyst in aqueous solution. Here, a heterogeneous catalyst is designed according to the acid sites required for the production of HMF from glucose. Then, pressure is reduced as a way to increase the production of HMF, and its mechanism is discussed. Moreover, the effects of temperature, time, and the recycling potential of catalysts are investigated.

## 2. Materials and Methods

### 2.1. Materials 

Glucose (99.5%), levoglucosan (99.0%), and furfural (99.5%) were purchased from Aladdin Reagent Co. Ltd. (Shanghai, China). HMF (99%, GR), ZrOCl_2_·8H_2_O (99.9%), levulinic acid (99.0%), formic acid (99.0%), and fructose (99.0%) were purchased from Macklin Reagent Co. Ltd. (Shanghai, China). NaOH (98%) was purchased from Fuchen Chemical Reagent Co. Ltd. (Tianjin, China). Boric acid (99.5%) was provided by Chinasun Specialty Products Co. Ltd. (Changshu, China). Sulfuric acid (98%) was supplied from Chemical Reagent Co. (Guangzhou, China).

### 2.2. Catalyst Preparation

We followed the typical approach of preparing a SO_4_^2−^/B_2_O_3_/ZrO_2_ catalyst by the coprecipitation and acidification method. ZrOCl_2_·8H_2_O and sodium hydroxide were dissolved in deionized water. The sodium hydroxide solution was added dropwise to the ZrOCl_2_·8H_2_O solution to adjust the pH to 9–10. Then, the precipitate was left to age for 5 h before separation and being washed with ethanol until no white precipitate was detected with 0.1 M AgNO_3_ (aq). The final sample was obtained after oven-drying at 110 °C for 6 h. After this, 0.7 M boric acid (aq) was added to the obtained ZrO(OH)_2_ with magnetic stirring over a period of approximately 6 h.

After five cycles of separation, washing and redispersion with ethanol, the powder obtained was oven-dried at 80 °C for 24 h and then calcined in air at 550 °C for 4 h. The obtained B_2_O_3_/ZrO_2_ was immersed in 1 M H_2_SO_4_ with stirring for 6 h. The suspension was then washed, dried and calcined as above.

### 2.3. Catalyst Characterization

The sample was ground for XRD. XRD spectra were recorded on a XRD-6000 X-ray diffractometer (PANalytical B.V., Almelo, The Netherlands) using Cu Kα radiation (40 kV voltage, 40 mA tube current). Diffraction patterns were recorded within a 2θ range from 20° to 60°. XPS spectra were recorded by an Axis Ultra DLD (Kratos Analytical Shimadzu Group Company, Manchester, UK) to detect the elements of Zr, S, B, and O. The data were calibrated by the C1s signal (284.4 eV). The sample was ground with KBr at the ratio of 1:100 and then pressed into disks for FTIR. FTIR spectra in the region of framework vibration (400–4000 cm^−1^) were recorded with a Tensor 27 infrared spectrometer (Bruker, Bremen, Germany). Acid properties were measured by temperature programmed desorption and pyridine infrared spectroscopy. Temperature programmed desorption of ammonia was performed on a AutoChem1 II 2920 (Micromeritics, Norcross, GA, USA). About 50 mg of catalyst was saturated with NH_3_ at 150 °C, flushed with He to remove physisorbed gas and then ramped to 650 °C at a heating rate of 10 °C/min under He flow. The pyridine infrared spectroscopy was determined by Nicolet 6700/TGAQ50 (Thermo Fisher Scientific, Waltham, MA, USA; TA instruments, New Castle, DE, USA). The catalyst was activated at 350 °C for 30 min. Pyridine was chemisorbed on the catalyst surface at 100 °C. Excess physisorbed pyridine was removed by holding the temperature at 100 °C for 15 min.

### 2.4. Conversion of Glucose to HMF

The experiment without pressure reduction was performed as follows: All reactions were carried out in 100 mL cylindrical stainless steel batch reactor (SLM 100, Beijing Century Sen Long experimental apparatus Co., Ltd., Beijing, China). The desired quantity of glucose (0.40 g), distilled water (40 mL), and catalyst (0.20 g) were introduced into the autoclave. Reactions were conducted at 150–190 °C with a reaction time range of 30–90 min in 30 min increments. This experiment was taken as the control. 

The experiment with pressure reduction was performed as follows: The reaction conditions are same as above. The steam was taken out by opening the vent valve slightly every 10 min when the reaction was ongoing (the pressure of the system was reduced due to the process, which is referred to as pressure reduction). It was condensed through the condensing tube for collection. For each experiment, a total of 15–25 mL of exact solution was taken out of the reactor, as shown in Table 1. The reactor monitored the reaction time, removed the vial from the heating board, and quenched the reaction in a cool board immediately after reaching the designed time. 

### 2.5. Product Analysis

Before the tests, the product was filtered with a 0.22 µm filter. The evaporated substance, products, and residual glucose were determined by Agilent 1100 liquid chromatography (HPLC). The specific conditions were as follows: chromatographic column, Bio-Rad Aminex HPX-87H column; detector, differential refractive detector (RID); mobile phase, 5 mM sulfuric acid; flow rate, 0.5 mL/min; column temperature, 50 °C; detector temperature, 40 °C. Figure 1 shows the HPLC chromatograph. The yield of products (HMF) and byproducts (levulinic acid, formic acid, levoglucosan, furfural), the conversion of glucose, extraction efficiency, and yield of humins were calculated by the external standard method. The formulas are as follows:(1)$$Yield\;of\;products\;in\;the\;solution\;\left(mol\%\right)=\frac{mol\;of\;product\;produced\;in\;solution}{mol\;of\;glucose\;initially\;charged}\times 100\%$$
(2)$$Conversion\;of\;glucose\;\left(mol\%\right)=\frac{mol\;of\;glucose\;reacted}{mol\;of\;glucose\;initially\;charged}\times 100\%$$
(3)electivety of products (mol%)=Yield of products×conversion of glucose
(4)Yield of humins (mol%)=(1−Selectivity of products)×Conversion of glucose×100%

## 3. Results

### 3.1. Catalyst Characterization

Figure 2a shows the XRD pattern of SO_4_^2−^/B_2_O_3_/ZrO_2_. From this, it can be seen that the XRD pattern of the catalysts exhibited characteristic diffraction peaks of tetragonal zirconia (t-ZrO_2_) and monoclinic zirconia (m-ZrO_2_). The typical diffraction peaks corresponded to the (111), (002), (200) crystal planes of t-ZrO_2_, which were consistent with the PDF card (JCPDS No. 414-0534). The following crystal planes, i.e., (110), (−111), (−102), (−112), (202), (220), and (−202), confirmed m-ZrO_2_, which were consistent with the PDF card (JCPDS No. 07-0343). There was no characteristic peak of B_2_O_3_ in the sample, indicating that B_2_O_3_ may be highly dispersed or amorphous [44].

Figure 2b shows the FT-IR spectra of SO_4_^2−^/B_2_O_3_/ZrO_2_. It can be seen from the figure that there were typical characteristic absorption peaks at 3403 cm^−1^, 1623 cm^−1^, 1450–1400 cm^−1^, 1150–1050 cm^−1^, 866 cm^−1^, and 501 cm^−1^. Among them, the bands at 1623 cm^−1^ and 3403 cm^−1^ were attributable to the O–H tensile vibration of adsorbed water on the surface and the H–O–H flexural vibration of bound water [45,46]. The band at 501 cm^−1^ belonged to the Zr–O vibration. The characteristic peaks at 1150–1050 cm^−1^ represented the formation of BO_4_ units [46], while the band at 1450–1400 cm^−1^ and 866 cm^−1^ represented the formation of BO_3_ units [44]. The absorption peak at 987–1137 cm^−1^ was attributed to the stretching vibration peak of S=O. Figure 2e shows the XPS of SO_4_^2−^/B_2_O_3_/ZrO_2_. The binding energies at 163.9, 179.7, and 527.1 eV corresponded to S 2p, B 1s, and O 1s, respectively. The Zr spectrum showed two peaks at 178.9 and 181.2 eV, which belonged to the Zr 3d5/2 and Zr 3d3/2 spin-orbitals of Zr^4+^, respectively. All the above results indicated that the required SO_4_^2−^/B_2_O_3_/ZrO_2_ was successfully prepared.

Generally, the adsorption of base molecules (e.g., pyridine or ammonia) combined with binding spectroscopy is a common technique, which was used to determine the nature of the surface active sites in solid acids [47]. The acidity on the surface of catalysts were determined by means of temperature programmed desorption (NH_3_-TPD) (shown in Figure 2c). The acid sites based on desorption temperature were divided into two categories: weak acid (150–250 °C), and medium acid (250–400 °C). The amount of surface acid was 1.78 mmol/g. Among them, the amount of weak acid and medium strong acid reached 0.74 mmol/g and 1.04 mmol/g, respectively. The acidic properties of SO_4_^2−^/B_2_O_3_/ZrO_2_ were investigated by FTIR spectra after pyridine chemisorption, as shown in Figure 2d. There were three bands in 1550–1400 cm^−1^. The peak at 1440 cm^−1^ indicated the presence of a Lewis acid. The one at 1530 cm^−1^ was attributed to the C-C stretching vibration of pyridium ion and has been used for the identification of desorption of a Brønsted acid. Another peak at 1512 cm^−1^ was assigned to the characteristic peak of the co-adsorption of pyridine molecules at the Brønsted acid sites and the Lewis acid sites [48,49].

### 3.2. Conversion of Glucose to HMF by SO^−^-/B_2_O_3_/ZrO_2_

Figure 3 showed the results that the influence of various temperature and time on the transformation of glucose to HMF over SO_4_^2−^/B_2_O_3_/ZrO_2_. The conversion of glucose was greatly affected by temperature. Glucose conversion increased from 45.9 to 60.3% with an increase in the temperature from 150 to 190 °C in 30 min. However, the production of HMF was not sensitive to reaction temperature: HMF yield increased little bit from 0.0 to 7.7%. The maximum yield of HMF reached 9.3% at 190 °C in 90 min when the conversion rate of glucose reached 64.6%. The reason for the low selectivity was that there are a large number of byproducts generated. With an increase of reaction time from 30 to 90 min, the HMF conversion increased slowly, but the conversion of glucose did not follow this trend. Previously, B_2_O_3_/ZrO_2_–Al_2_O_2_ containing Brønsted and Lewis acid sites was conducted to produce HMF in DMSO [44]. The yield was 41.2%, much higher than that of SO_4_^2-^/B_2_O_3_/ZrO_2._ The main reason was that many side-reactions may happen in water, while HMF was more stable in DMSO [50].

According to previous reports [51] (as shown in the Figure 4), the side-reaction included into four paths: (1) isomerization of glucose to fructose; (2) dehydration of glucose to levoglucosan (LGA), furfural (FF); (3) hydration of HMF to levulinic acid (LA) and formic acid (FA); and (4) condensation of HMF to humins. Soluble byproducts such as LGA, FF, LA, FA, and fructose can be detected by HPLC. It turns out that in addition to unreacted glucose and HMF, there was a small amount of fructose that was less than 10%. As can be seen in Table 2, other byproducts were thought to be insoluble humins produced by HMF condensation, where the yield was 46.2%. However, there is no means to qualitatively analyze the main components of humins.

### 3.3. Conversion of Glucose to HMF by SO_4_^2−^/B_2_O_3_/ZrO_2_ with Pressure Reduction

#### 3.3.1. The Possible Mechanism of Pressure Reduction

According to Table 2, HMF was not detected by HPLC in the vapor, which meant that HMF remained in the liquid product in a high content. Compared to the experimental pressure with vapor pressure (as can be seen in Figure 5), the phenomenon that only water was evaporated can be explained. If the extracted vapor was 1 or 2 mL, the temperature and pressure in the kettle would reduce due to the removal of steam. Obviously, the experimental pressure was between the saturated vapor pressure of water and HMF. In other words, releasing vapor pressure could promote water to boil. 

#### 3.3.2. Conversion of Glucose to HMF with Pressure Reduction

As can be seen in the Figure 6, the reaction temperature had a certain influence on the yield of HMF and the conversion of glucose with reducing the pressure. As temperature rose, the glucose conversion gradually increased. For example, it rose from 36.1 to 67.3% rapidly when the temperature elevated from 150 to 190 °C in 30 min, which indicates that a high reaction temperature could improve the mass transfer and heat transfer between the substrate and the catalyst. The acid sites of the catalyst in the reaction system contacted with the substrate more fully, which promoted the conversion of glucose. The trend of HMF yield was consistent with glucose conversion when it reacted for 30 min and 60 min. At 90 min, however, the yield of HMF increased and then decreased. For example, the yield of HMF was 4.2% at 150 °C in 30 min. When the temperature rose to 190 °C, the yield of HMF increased to 31.5%. With the reaction time increasing to 90 min at 190 °C, the yield of HMF decreased to 24.0%. The maximum yield reached within 60 min at 190 °C was 35.2%. These results indicate that the formation of polymers species hinder the active sites with prolonging the reaction time. The byproducts in the reaction solution were also quantified. Although the catalyst prepared for the selective production of HMF was not very effective, this method of pressure reduction greatly improved its yield. The result was better than that of HCl [43], ZnCl_2_ [52], and cellulose-derived carbonaceous [53] in the aqueous phase. Therefore, this method can be widely used for the production of HMF in the aqueous phase on a solid catalyst. 

The occurrence of side reactions is inevitable in water. However, the soluble byproducts such as LGA, FF, LA, FA, and fructose were not detected by HPLC, as shown in Table 3. Nevertheless, the yield of humin decreased from 46.2 to 36.3%. The indexes were much better than that without pressure reduction, which indicates that pressure reduction can effectively promoted the production of HMF. To this extent, by reducing pressure, it not only improved the yield of HMF and the conversion of fructose, but it also reduced the condensation products of HMF. Based on this result, the method of pressure reduction have remarkably unique and environmentally friendly benefits, including that the fact that the isolation and purification of intermediate compounds can be avoided. Firstly, the catalyst and humins were filtrated by 0.22 µm. Since there are no other byproducts (e.g., LA, FA, LGA, FF, and fructose), HMF only need to be purified by distillation from unreacted glucose and water. The HMF and water are distillation from glucose. Finally, the purified HMF is obtained from the solution above through a rotary evaporator.

Zhou’s research [34] showed that the low boiling solvent produced bubbles when boiling: bubbles rose from the bottom to the upper area, promoting the upward movement of HMF to ionic liquid for inhibiting its side reactions; this process also causes the reaction system to be agitated, promoting the ionic liquid with high viscosity to mass transfer. In this paper, the main reason for using pressure reduction to promote the dehydration of glucose to HMF is that pressure reduction promoted the boiling of water caused by the reduced pressure (as shown in Figure 7). The bubbles produced by boiling causes the disturbance of the reaction system, which promoted the rapid desorption of HMF from the acid sites of solid catalyst and further promoted the adsorption of glucose with more acid sites, which improves the conversion of glucose and the yield of HMF. Both methods are to promote the production of HMF by solvent boiling, and the yield of HMF is increased by more than three times. However, the methods used in this study are more environmentally friendly and economical. This is an efficient method to produce HMF by avoiding the use of organic solvents.

### 3.4. Reuse Potential of Catalysts

The stability of SO_4_^2−^−B_2_O_3_/ZrO_2_ catalyst was investigated with pressure reduction. The results are shown in Figure 8. In the second reuse, HMF yield decreased slightly from 35.2 to 34.1%. In the third reuse, the yield of HMF decreased to 24.5%, which may be due to the pore structure of the catalyst, which blocked byproduct humus and other solid substances.

## 4. Conclusions

The method of using pressure reduction in the procedure of catalytic conversion with solid acid in aqueous system is an effective strategy to increase HMF yield. The possible mechanism is to use the bubbles produced by boiling, as these bubbles can cause turbulence in the solution, in turn promoting the diffusion and transfer of HMF from the solid catalyst and avoiding further side reactions at acid sites. Under the same conditions, the yield of HMF can be increased from 9.3 to 35.2% (at 190 °C for 60 min) when reducing pressure. In addition, the SO_4_^2−^−B_2_O_3_/ZrO_2_ catalysts can also keep stable, and the yield of HMF with the third recycling of this catalyst can also reach 24.5%.

## Figures and Tables

**Figure 1 polymers-13-02096-f001:**
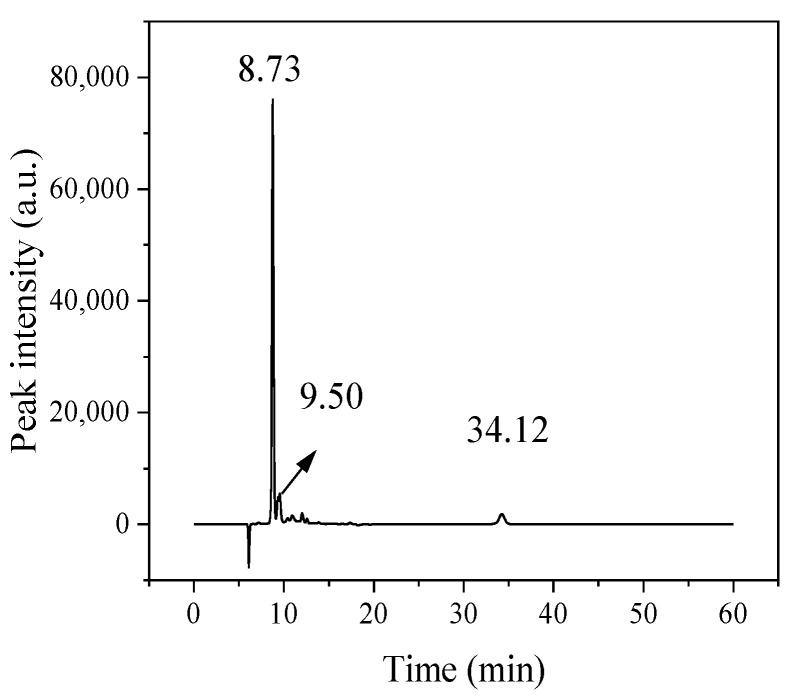
HPLC chromatograph (8.73 min: glucose; 9.50 min: fructose; 34.12 min: HMF).

**Figure 2 polymers-13-02096-f002:**
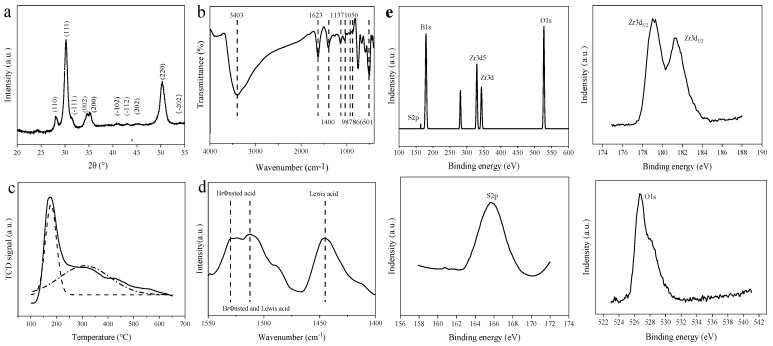
Acid properties and structure characterization of catalysts. (**a**) XRD pattern; (**b**) FTIR spectra; (**c**) temperature programmed desorption; (**d**) FTIR spectra of pyridine chemisorption; (**e**) XPS pattern.

**Figure 3 polymers-13-02096-f003:**
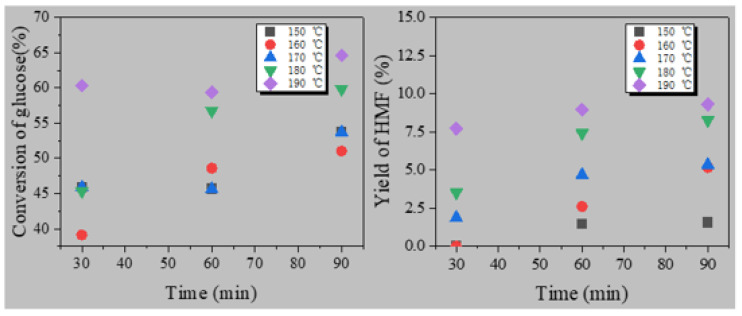
The effect of temperature and time on conversion of glucose to HMF over SO_4_^2−^/B_2_O_3_/ZrO_2_.

**Figure 4 polymers-13-02096-f004:**
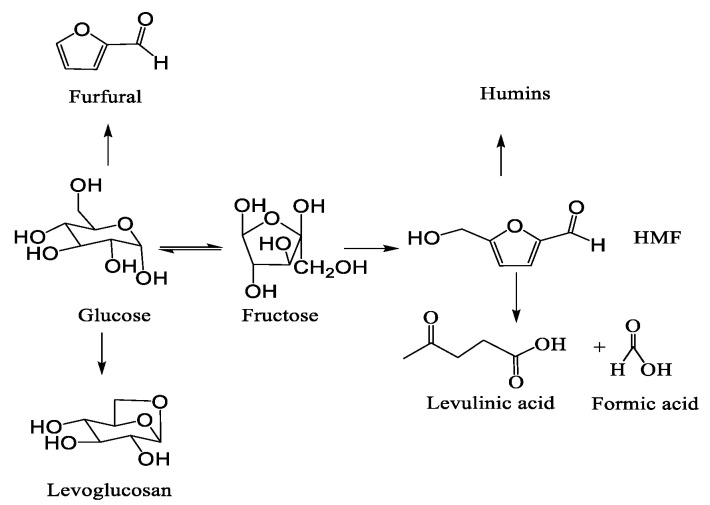
Reaction pathway of glucose.

**Figure 5 polymers-13-02096-f005:**
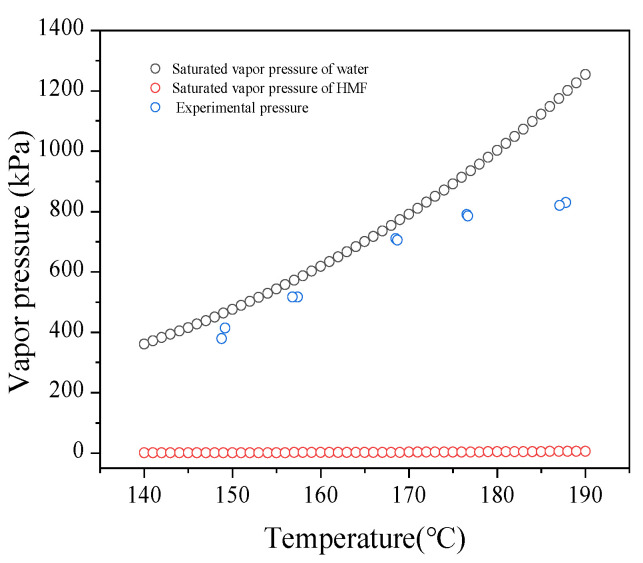
Saturated vapor pressure of water and HMF at different temperatures.

**Figure 6 polymers-13-02096-f006:**
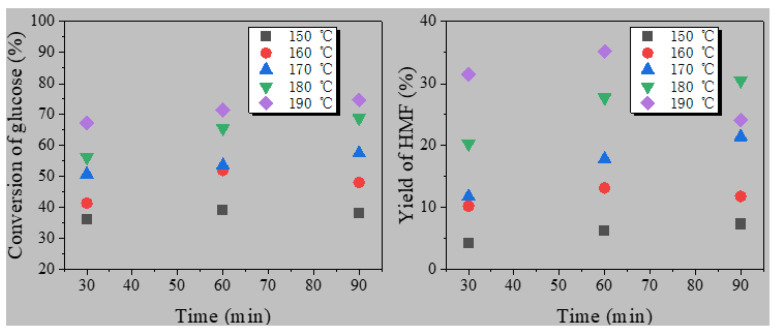
The effect of temperature and time on conversion of glucose to HMF with pressure reduction over SO_4_^2−^/B_2_O_3_/ZrO_2_.

**Figure 7 polymers-13-02096-f007:**
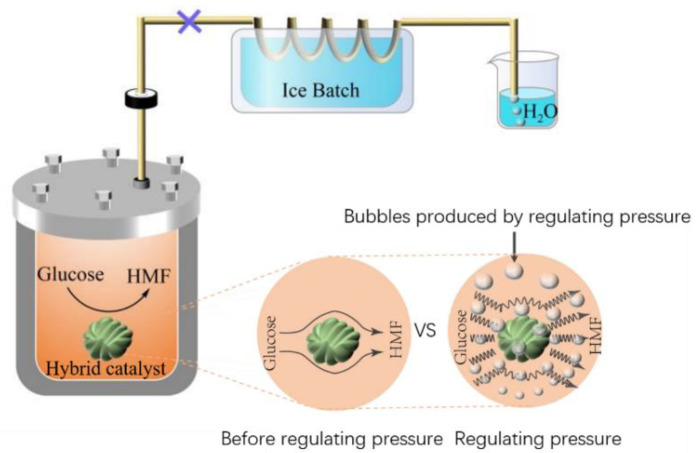
Mechanism of pressure reduction to enhance catalytic efficiency.

**Figure 8 polymers-13-02096-f008:**
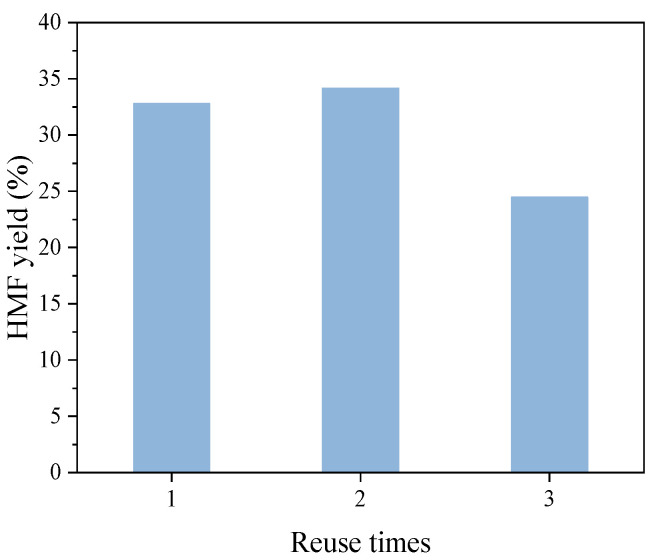
Reuse potential of SO_4_^2−^−B_2_O_3_/ZrO_2_: 0.40 g glucose, 0.20 g catalysts, 40 mL water, at 190 °C, for 60 min.

**Table 1 polymers-13-02096-t001:** The volume of solution in the vessel.

Temperature (°C)	Time (min)	Volume (mL)
150	30	23
	60	21.5
	90	25
160	30	21
	60	17
	90	22
170	30	22
	60	22
	90	22.5
180	30	21
	60	20
	90	20
190	30	20
	60	21.5
	90	22.5

**Table 2 polymers-13-02096-t002:** Concentration of HMF in the vapor.

Temperature (°C)	Time (min)	Yield of HMF (%)
150	30	0.062
	60	0.069
	90	0.075
160	30	0.100
	60	0.152
	90	0.150
170	30	0.140
	60	0.185
	90	0.243
180	30	0.239
	60	0.284
	90	0.336
190	30	0.405
	60	0.447
	90	0.389

**Table 3 polymers-13-02096-t003:** The byproducts in the solution at a temperature of 190 °C and time of 60 min.

Controls-Yield (%)						Pressure Reduction -Yield (%)					
Soluble Products					Insoluble Products	Soluble Products					Insoluble Products
LA	FA	LGA	FF	Fructose	Humins	LA	FA	LGA	FF	Fructose	Humins
0.0	0.0	0.0	0.0	0.0	36.3	0.0	0.0	0.0	0.0	4.2	46.2

## Data Availability

The data presented in this study are available on request from the corresponding author. The data are not publicly available.

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
