# Peer review of "Pressure Reduction Enhancing the Production of 5-Hydroxymethylfurfural from Glucose in Aqueous Phase Catalysis System"

_polymers, 2021, doi:10.3390/polym13132096_

Round 1
Reviewer 1 Report
The manuscript presented for evaluation, titled 'Regulating Pressure: Enhancing the Production...' by Bo Li et al., describes a kind of novel strategy to improve the yield of the synthesis of hydroxymethylfurfural (HMF) in water using a practical approach of heterogeneous catalysts. The work is in generally well performed, however there are some point that are difficult to be understand; some typos are also present. The method used for characterization of the catalyst are in accordance with the literature. The idea of the pressure regulation to improve the yields of HMF seems to have some impact on the results. The paper has 40 References written in the correct format. I would suggest to the authors to include also some details about the mechanism of formation of HMF from glucose. There are also some minor corrections that should be addressed before publication:
-line 1- type of paper should be mentioned;
-Abstract can be improved;
-First paragraph in Section 2.4 is not clear;
-lines 207 and 208-is not clear, not understandable;
-Chapter 4 (Discussion) has only 3 line, this should be more developed.
In conclusion, the work can be accepted for publication, but needs some improvements, the most important being i) make the manuscript better understandable and ii) add more Discussion in Chapter 4.
Author Response
Response to Reviewer 1 Comments
Point 1: line 1- type of paper should be mentioned.
Response 1: Thanks! The type has been mentioned in the revised manuscript.
Point 2: Abstract can be improved.
Response 2: Abstract has been rewritten in the revised manuscript according to the reviewer’s advice.
Point 3: First paragraph in Section 2.4 is not clear;
Response 3: Thanks for suggestion. The first paragraph in Section 2.4 has been reinterpreted.
Point 4: lines 207 and 208-is not clear, not understandable.
Response 4: Thanks. The related content of this section has been revised in the revised manuscript.
Point 5: Chapter 4 (Discussion) has only 3 line, this should be more developed.
Response 5: Discussion has been added to the revised manuscript according to the reviewer’s advice.

Reviewer 2 Report
Dear editor, in the present work HMF is produced from glucose. Even thought the subject is very interesting I do not think that this work is appropriate for publication in Polymers in the present form and lot of changes and additional text should be added from authors in order to improve it.
It was reported in introduction that: 5-hydroxymethylfurfural (HMF) regarded as "Sleeping Giant", plays an important role in platform compounds. However, there are no mentioned applications concerning the produced polymers. Some examples could be mentioned to be more completed this part. For example, please see Eur. Polym. J. 83; 202–229: 2016 and Polymers 12, 1209; 2020, http://doi:10.3390/polym12061209. Other works can be also added.
The results have been presented in oversimplified way and discussion is missing as well as comparison with other previous reported works. The whole references have been mentioned in introduction. It seems that authors have discover for first time these catalysts and the conversion of glucose to HMF. I do not think that this work can be published as it is and substantially revision will be needed.
What are the benefits of using these catalysts and proposed procedures with the already mentioned in literature? Are any in yield, conversion, etc.
What are the other byproducts were not detected by HPLC? A table for each case could be mentioned.
How can separated HMF from the other byproducts? What is the purity of produced HMF?
Please remove this short discussion part. It has no meaning and results, and discussion should be included in one part. The discussion should be extended.
In Abstract or mainly in conclusions the optimum conditions for HMF production of this study should be mentioned.
Author Response
Response to Reviewer 2 Comments
Point 1: It was reported in introduction that: 5-hydroxymethylfurfural (HMF) regarded as "Sleeping Giant", plays an important role in platform compounds. However, there are no mentioned applications concerning the produced polymers. Some examples could be mentioned to be more completed this part. For example, please see Eur. Polym. J. 83; 202–229: 2016 and Polymers 12, 1209; 2020, http://doi:10.3390/polym12061209. Other works can be also added.
Response 1: Thanks for the suggestion. HMF can be produced from biomass, cellulose. But due to the complexity of the dehydration process, the yield of HMF is low. So, to reduce the difficulty of production, glucose which is hydrolysis product of cellulose was used to produce HMF. The method of regulating pressure reported in this paper has a positive effect on the turbulence of the reaction system, which also provides reference value for more complex polymers to prepare HMF. In addition, FDCA, oxidation product of HMF, can prepare a plasticizer, polyester and other polymer materials. Recommended literatures and more information about polymers have been added in the introduction.
Point 2: The results have been presented in oversimplified way and discussion is missing as well as comparison with other previous reported works. The whole references have been mentioned in introduction. It seems that authors have discover for first time these catalysts and the conversion of glucose to HMF. I do not think that this work can be published as it is and substantially revision will be needed.
Response 2: We have revised the results and discussions. The catalyst which possesses BrΦnsted and Lewis acid sites in this work are not prepared for the first time, but a common way to prepare HMF in glucose aqueous solution. However, this paper is not focus on how the catalyst can performed. The point of this paper is to develop a new method-regulating pressure to improve the yield of HMF on the basis of catalyst. Compare with other previous reported works, yield of HMF is much higher which indicates the method makes the current stagnant progress of HMF industrialized. Therefore, the work of this paper is very meaningful, published in the journal will provide a reference for readers in the field.
Point 3: What are the benefits of using these catalysts and proposed procedures with the already mentioned in literature? Are any in yield, conversion, etc.
Response 3: The catalyst prepared in this paper is universal for production of HMF. But regulating pressure is a new method to improve the yield of HMF based on catalyst.
Point 4: What are the other byproducts were not detected by HPLC? A table for each case could be mentioned.
Response 4: According to previous reports, the side reaction included into three paths: (1) dehydration of glucose to furfural, levoglucosan, fructose; (2) hydration of HMF to levulininc acid and formic acid; (3) depolymerization of HMF to humins. The byproducts in paths (1) and (2) were detected by HPLC and they are little concentration in the products. Humins produced in path (3) are complex substances, which can not be detected by existing means. But, the yield of humins was calculated by the yield, selectivity of HMF and conversion of glucose.
Point 5: How can separated HMF from the other byproducts? What is the purity of produced HMF?
Response 5: The reaction system is heterogeneous catalytic systems in water. HMF can be separated from solid products and catalyst by filtration. Since there are no by-products such as levulinic acid and formic acid in the products, HMF only needed to be separated from water and unreacted glucose. Theoretically, HMF can be purified by atmospheric distillation and vacuum distillation. But it belongs to the subsequent reprocessing of HMF which is not the point of this paper.
Point 6: Please remove this short discussion part. It has no meaning and results, and discussion should be included in one part. The discussion should be extended.
Response 6: The Discussion has been extended, including the results and how they can be interpreted from the perspective of previous studies and of the working hypotheses. Moreover, the vision of future for production of HMF from other materials has been appended.
Point 7: In Abstract or mainly in conclusions the optimum conditions for HMF production of this study should be mentioned.
Response 7: The maximum yield of HMF of this study was achieved at 190 ℃ in 60 min. This content was reported in the section of Results in original manuscript, and has been added In Abstract and Conclusion in revise manuscript according to the reviewer’s advice.

Reviewer 3 Report
5-Hydroxymethylfurfural (HMF) is a promising platform chemical for the production of value-added chemicals and fuels. The production of such chemicals from renewable biomass is experiencing enormous interest recently due to the latest developments in sustainable production processes and CO2 reductions. Conversion of abundantly available and cheap glucose for HMF synthesis seems to be systematically investigated in the submitted manuscript while controlling the pressure in a green water system. Transforming biomass HMF into high-value chemical substances is a significant area of research that reduces the emissions. Manuscript is well presented with supportive physical and chemical validations, however, some revision is required before rendering a final decision.
- Title : “Regulating Pressure” needs to be reconsidered.
introduction
- Section 1 (lines 43 – 37) The major concern/obstacle with employing 5-HMF normally involves the isomerization of glucose to fructose and dehydration of fructose to 5-HMF. This may lead to the catalyst requirement of both Lewis acid and Brønsted acid for isomerization and dehydration. How this has been addressed (or overcome) in the submitted work?
- Section 1: What is FCDA (line 38)?
- The disadvantages of using organic solvents as media for glucose conversion can be compared in the introduction so that readers can get the merit of the currently used aqueous phase.
- The main factors (solvent amount, and catalyst concentration; in addition to temperature and reaction time shown) affecting the yield and selectivity of 5-HMF can be briefly touched.
Experimental
- How the extraction efficiency been calculated
- Section 2.3; is it 40 mA (or) 30 mA tube current for XRD?
- Why pyridine and what is its role?
- How are samples prepared for XRD and FTIR?
- What is the “RID” deductor (line 137)?
- Unable to see any HPLC chromatograms results but stated in this section
Results and Discussion
- Section 3.1 (lines 150-151) XRD – “There was no characteristic peak of B2O3 in the sample, indicating that B2O3 may be highly dispersed or amorphous.” Could this be due to the insignificant amount of B2O3 (below the detectable limit of XRD?) Please interpret the XRD data reported in the literature (Solid State Ionics 179 (2008) 355) for B2O3
- Section 3.1; second paragraph: The IR peak assignments made on O-H and C – C stretching vibrations need to refer to the relevant literature (such as Electrochem. Solid-State Letters 14 (2011) A86; and ACS Applied Energy Mater. 3 (2020) 12385).
- Figure caption for 1(e) XPS spectra need to be detailed for which elements, the analysis was performed?
- HMF synthesis from constituent monosaccharides of sucrose and fructose can be compared.
- Since the product obtained from the synthesis was contaminated with catalyst, does the purification of 5-HMF is required?
- From Figs 3 – 5 what is the optimised value for glucose conversion to HMF with high yield, please specify.
- Figures 3 – 4 axes label what does it mean “with” and without”? unclear.
Author Response
Response to Reviewer 3 Comments
Point 1: Title: “Regulating Pressure” needs to be reconsidered.
Response 1: We have changed “Regulating Pressure” to “Reducing pressure” in the revised manuscript.
Point 2: Section 1 (lines 43 – 37) The major concern/obstacle with employing 5-HMF normally involves the isomerization of glucose to fructose and dehydration of fructose to 5-HMF. This may lead to the catalyst requirement of both Lewis acid and Brønsted acid for isomerization and dehydration. How this has been addressed (or overcome) in the submitted work?
Response 2: The catalysts were prepared by impregnated sulfuric acid and coupled zirconia with boron oxide, which made the catalysts have Lewis acid sites and BrΦnsted acid sites. The presence of these two acid sites has been confirmed in Figure 2 (d). The experiment has also been described. The details were specified in Response 8.
Point 3: What is FCDA (line 38)?
Response 3: FDCA is 2,5-Furandicarboxylicacid, which has been revised and explained in the Introduction.
Point 4: The disadvantages of using organic solvents as media for glucose conversion can be compared in the introduction so that readers can get the merit of the currently used aqueous phase.
Response 4: Thanks for the suggestion. The disadvantages of using organic solvents as media has been added in the revised manuscript (line 49-50).
Point 5: The main factors (solvent amount, and catalyst concentration; in addition to temperature and reaction time shown) affecting the yield and selectivity of 5-HMF can be briefly touched.
Response 5: Yes, many factors do affect the reaction and the results. But this work focuses on the effect of pressure variations on the procedure of catalytic reaction. Subsequently, the optimization experiments will be conducted with these factors mentioned by reviewer, which will be a great significance for industrialization.
Point 6: How the extraction efficiency been calculated
Response 6: The extraction efficiency was calculated by the formula:
.
The formula and data have been added in the section 2.5 (line 149) and Table 2 (line 224).
Point 7: Section 2.3; is it 40 mA (or) 30 mA tube current for XRD?
Response 7: It is 40 mA tube current for XRD.
Point 8: Why pyridine and what is its role?
Response 8: Generally, the adsorption of base molecules (e.g. pyridine or ammonia) combined with binding spectroscopy is a common technique, which was used to determine the nature of the surface active sites in solid acids[1]. Pyridine is characterized by strong adsorption with stable aromatic structure [2]. There are peaks when pyridine adsorbed with acid sites at different wavenumbers, wherein the peaks at 1450 cm-1 and 1540 cm-1 can be attributed to Lewis acid sites and BrΦnsted acid[3]. Therefore, Pyridine is as a molecular probe for identify the solid catalyst surface acid sites Relevant content and references (lines 177-179) have been added to the revised manuscript.
References:
- Yin, F.; Blumenfeld, A.L.; Gruver, V.; Fripiat, J.J. NH3 as a probe molecule for NMR and IR study of zeolite catalyst acidity. J Phys Chem B 1997, 101, 1824-1830, doi:DOI 10.1021/jp9618542.
- Alberici, R.M.; Jardim, W.E. Photocatalytic destruction of VOCs in the gas-phase using titanium dioxide. Appl Catal B-Environ 1997, 14, 55-68, doi:Doi 10.1016/S0926-3373(97)00012-X.
- Corma, A. From microporous to mesoporous molecular sieve materials and their use in catalysis. Chem Rev 1997, 97, 2373-2419, doi:DOI 10.1021/cr960406n.
Point 9: How are samples prepared for XRD and FTIR?
Response 9: The sample was ground for XRD. The sample was ground with KBr at the ratio of 1:100, then pressed into disks for FTIR. We have added the details in lines 105 and 109-110.
Point 10: What is the “RID” deductor (line 137)?
Response 10: RID is differential refractive detector, a universal detector for HPLC. We added the full name of RID in the revised manuscript.
Point 11: Unable to see any HPLC chromatograms results but stated in this section
Response 11: We have added the chromatograms results (lines 151-152) in the section 2.5 according to reviewer’s advice.
Point 12: Section 3.1 (lines 150-151) XRD – “There was no characteristic peak of B2O3 in the sample, indicating that B2O3 may be highly dispersed or amorphous.” Could this be due to the insignificant amount of B2O3 (below the detectable limit of XRD?) Please interpret the XRD data reported in the literature (Solid State Ionics 179 (2008) 355) for B2O3
Response 12: The strong intensity of B 1S peak in XPS spectrum indicated the high content of B2O3 in the catalyst. The preparation method of the catalyst in this article is modified based on the preparation method of the catalyst in the previous literature (Waste and Biomass Valorization, 2018, 9(11):2181-2190). This literature showed that B2O3 in the catalyst was amorphous and dispersed. Therefore, we believe that B2O3 in the catalyst is also amorphous and dispersed. We have revised the relevant contents and added the reference in the revised manuscript.
Point 13: Section 3.1; second paragraph: The IR peak assignments made on O-H and C – C stretching vibrations need to refer to the relevant literature (such as Electrochem. Solid-State Letters 14 (2011) A86; and ACS Applied Energy Mater. 3 (2020) 12385).
Response 13: Thanks for the suggestion. We have supplemented the relevant literatures in the revised manuscript (lines 168 and 190).
Point 14: Figure caption for 1(e) XPS spectra need to be detailed for which elements, the analysis was performed?
Response 14: Ok. More elements and the analysis have been supplemented in line 108.
Point 15: HMF synthesis from constituent monosaccharides of sucrose and fructose can be compared.
Response 15: Thanks for the suggestion. The relevant literature has been supplemented and a brief comparison and discussion have been added (lines 39-41) in the revised manuscript.
Point 16: Since the product obtained from the synthesis was contaminated with catalyst, does the purification of 5-HMF is required?
Response 16: In the system, HMF was dissolved in water and the catalyst is in a solid state. Therefore, it is easy to separate HMF from catalyst, just like filtering. The relevant content (lines 264-170) has been added in the revised manuscript.
Point 17: From Figs 3 – 5 what is the optimised value for glucose conversion to HMF with high yield, please specify.
Response 17: The optimal conditions were at 190 ℃ for 60 min which leaded the yield of HMF to 35.2% with reducing pressure. But it reached the maximum yield (9.3%) at 190 ℃ for 90 min in the controls sample. This result has been added to the parts of abstract, results and conclusions.
Point 18: Figures 3 – 4 axes label what does it mean “with” and without”? unclear.
Response 18: “with” means with reducing pressure; “without” means controls. We have changed the part into two groups. The details have been showed in Figure 3 and Figure 6.

Round 2
Reviewer 2 Report
Dear editor,
In the revised manuscript most of the proposed comments have been ignored. Few additions have been done in introduction and results-discussion part. The most important of this study, which is the produced byproducts during this reaction (I propose to add a table with these), their percentage, how can separate HMF from the other byproducts, the purity of produced HFM, etc., have been not addressed in the paper. I proposed to do a substantial revision, but the authors tried just to response to my comments!!! The research is incomplete with many missing information. Lot of parts need further clarification and thus additional experiments may needed. For all these I do not feel that the paper can by published in Polymers.
Author Response
Response to Reviewer 2 Comments
Point 1: It was reported in introduction that: 5-hydroxymethylfurfural (HMF) regarded as "Sleeping Giant", plays an important role in platform compounds. However, there are no mentioned applications concerning the produced polymers. Some examples could be mentioned to be more completed this part. For example, please see Eur. Polym. J. 83; 202–229: 2016 and Polymers 12, 1209; 2020, http://doi:10.3390/polym12061209. Other works can be also added.
Response 1: Furan-2,5-dicarboxylic acid (FDCA), oxidation product of 5-hydroxymethlfurfural (HMF), is a monomer for the production of the biodegradable plastic (Polyethylene 2,5-furandicarboxylate, PEF), which can replace the non- degradable plastics, such as PET. Recommended literatures and more information about polymers have been supplemented in the first paragraph in Introduction.
Point 2: The results have been presented in oversimplified way and discussion is missing as well as comparison with other previous reported works. The whole references have been mentioned in introduction. It seems that authors have discover for first time these catalysts and the conversion of glucose to HMF. I do not think that this work can be published as it is and substantially revision will be needed.
Response 2: We have to emphasize that the work does not focus on how the catalyst can performed. Our focus is to develop a new method with reducing pressure to improve the yield of HMF on the basis of catalyst. Compare with other previous reported works, the yield of HMF in this paper is much higher than these values, which indicates the method might improve or accelerate the industrialization of HMF. Therefore, our work of this paper should be meaningful. If it is published in this journal, it will provide a very useful reference for readers in the field.
The results and discussions have been extended and supplemented.
Point 3: What are the benefits of using these catalysts and proposed procedures with the already mentioned in literature? Are any in yield, conversion, etc.
Response 3: The catalyst prepared in this paper is universal for production of HMF, which is not the focus of this work. Our work focused on finding an effective method, reducing pressure, to improve the yield of HMF based on catalyst. This method improves the production on SO42-/B2O3/ZrO2 and its performance is better than homogeneous catalysts like HCl and ZnCl2. Moreover, comparisons with other previous reported works have been added in the section 3.3.2 (line 253-267).
Point 4: What are the other byproducts were not detected by HPLC? A table for each case could be mentioned.
Response 4: Humins cannot be detected by HPLC, but other byproducts (such as fructose, levulinic acid, formic acid, levoglucosan and furfural) can be detected by HPLC. The details of all by-products as above mentioned has been shown in Table 3 in the revised manuscript. The details can be found in line 202. In addition, the yield of byproducts and humins can be calculated by Equation (3), (4) and (5).
Point 5: How can separated HMF from the other byproducts? What is the purity of produced HMF?
Response 5: The reaction system is heterogeneous catalytic systems in water. HMF can be separated from solid products (humins) and catalyst by filtration. Since there are no by-products such as fructose, levulinic acid, formic acid, levoglucosan and furfural in the products, HMF only needed to be separated from water and glucose. Theoretically, HMF can be purified by atmospheric distillation and vacuum distillation. But it belongs to the subsequent reprocessing of HMF and it is not the focus of this paper. We will study it in the following work. Therefore, we have added a brief explanation of HMF purification in section 3.3 (lines 263-270) of the revised manuscript for further reference.
Point 6: Please remove this short discussion part. It has no meaning and results, and discussion should be included in one part. The discussion should be extended.
Response 6: Ok. This short discussion part has been deleted. And the discussion has been extended in section 3.3 for a full discussion
Point 7: In Abstract or mainly in conclusions the optimum conditions for HMF production of this study should be mentioned.
Response 7: Thank for the suggestion. It has been supplemented in abstract and conclusion in revised manuscript.
